# A Review of Immunotherapeutic Strategies in Canine Malignant Melanoma

**DOI:** 10.3390/vetsci6010015

**Published:** 2019-02-12

**Authors:** Ramón M. Almela, Agustina Ansón

**Affiliations:** Department of Clinical Sciences, Cummings School of Veterinary Medicine, Tufts University, Grafton, MA 01536, USA; agusanson@um.es

**Keywords:** melanoma, malignant, canine, immunotherapy, vaccine, gene therapy, review

## Abstract

In dogs, melanomas are relatively common tumors and the most common form of oral malignancy. Biological behavior is highly variable, usually aggressive, and frequently metastatic, with reported survival times of three months for oral or mucosal melanomas in advanced disease stages. Classical clinical management remains challenging; thus, novel and more efficacious treatment strategies are needed. Evidence-based medicine supports the role of the immune system to treat neoplastic diseases. Besides, immunotherapy offers the possibility of a precise medicinal approach to treat cancer. In recent years, multiple immunotherapeutic strategies have been developed, and are now recognized as a pillar of treatment. In addition, dogs represent a good model for translational medicine purposes. This review will cover the most relevant immunotherapeutic strategies for the treatment of canine malignant melanoma, divided among five different categories, namely, monoclonal antibodies, nonspecific immunotherapy activated by bacteria, vaccines, gene therapy, and lymphokine-activated killer cell therapy.

## 1. Introduction

Melanoma is a relatively common cancer of dogs arising from melanocytes. It accounts for 3% of all neoplasms and up to 7% of all malignant tumors, and it is the most common oral malignancy (56%). Primary melanomas can also occur in the nail bed, footpad, eye, gastrointestinal tract, and mucocutaneous junction [1,2]. The biological behavior of canine malignant melanoma is extremely variable and best characterized by anatomic site, size, stage, and histologic parameters. The anatomic site of melanoma is highly, although not completely, predictive of local invasiveness and metastatic propensity. Melanomas involving haired-skin that is not in proximity to mucosal margins often behave in a benign manner, whereas oral or mucosal melanomas are often malignant and metastatic with a reported median survival time (MST) of 14 months for stage I and three months for stage III disease. Size is evaluated in the staging of dogs with melanoma along with lymph node involvement and distant metastasis. For instance, tumors <2 cm in diameter and without evidence of metastasis are classified as stage I with a reported MST of up to 19 months. On the other hand, tumors between 2–4 cm in diameter without evidence of metastasis are classified as stage II and have a reported MST of up to six months (Table 1). Histologic parameters such as Ki67 expression, a proliferation index histologic marker, can also predict behavior as in canine cutaneous melanoma. An increased Ki67 expression correlates with a moderate-to-high metastatic propensity [2]. Classical clinical management includes radical surgical excision [3], radiation therapy, either as primary or adjuvant therapy, and a combination of chemotherapy and radiation therapy [2]. These classical therapies remain a challenge to control disease. 

It is increasingly recognized that the immune system plays a critical role in the development and progression of cancer [4]. The treatment of cancer has largely relied on killing tumor cells with chemotherapy and radiotherapy. This approach, however, has limitations, including severe systemic toxicities, bystander effects on normal cells, recurrence of drug-resistant tumor cells, and the inability to target micrometastases or subclinical disease [5]. In addition, there is no clear evidence that chemotherapy extends MSTs and decreases local recurrence or metastasis risks [2]. Thus, the pursuit of more efficacious and tolerable treatments, in conjunction with an increased understanding of cancer immunology, spurred intense research in the development of immunotherapies directed against cancer [5]. These immunotherapies have emerged as an interesting approach to managing canine malignant melanoma (CMM), and multiple immunotherapeutic strategies have been developed targeting the innate and/or adaptive arms of the immune system. As a result, newer approaches to systemic management have been developed—for instance, gene therapy [6], lymphokine-activated killer (LAK) cell therapy [7], and monoclonal antibodies [8]. These immunotherapies seem, in some cases, to improve survival times. 

The aim of this review is to summarize the current and most relevant immunotherapeutic strategies for the treatment of CMM, divided among five different categories: monoclonal antibodies, nonspecific immunotherapy activated by bacteria, vaccines, gene therapy, and lymphokine-activated killer cell therapy. Studies performing at least two approaches fell in one category, chosen at the discretion of the author. This review focuses on the most relevant English-written, peer-reviewed papers of preclinical experiments and clinical data collected from client-owned dogs with spontaneously arising tumors. The search was based on PubMed, Web of Science, and Scopus using the Boolean chains “dog” OR “dogs” OR “canine” AND “melanoma” AND “immunotherapy” OR “vaccine”. The search was performed and revised within the time frame from October the 1st to December the 15th of 2018 and retrieved a total of 241 papers. Irrelevant, duplicated, or beyond-the-scope papers were excluded. 63 papers were included for the review.

## 2. Monoclonal Antibodies

### 2.1. Checkpoint Inhibitors

It is now recognized that established tumors have numerous mechanisms for suppressing the antitumor immune response, including production of inhibitory cytokines, recruitment of immunosuppressive immune cells, and upregulation of coinhibitory receptors known as immune checkpoints. The primary effector cells of the adaptive immune response against cancer are the T lymphocytes. Importantly, T-cell priming/activation is tightly regulated by costimulatory or coinhibitory signals known as immune checkpoints [4] that provide a regulatory feedback mechanism to limit the effector phase of T-cell expansion and function. They play key parts in the tolerance for self-antigens and provide the basis for fine-tuning the T-cell response [9]. These inhibitory pathways are upregulated in many cancers, and immune checkpoints play critical roles in cancer-associated immune suppression and immune evasion. Increased understanding of these mechanisms has led to the development of immunotherapies targeting cancer-associated immunosuppression [4]. Currently, the most salient modality is the use of targeted monoclonal antibodies (mAbs) against regulatory immune checkpoint molecules which has revolutionized the treatment of cancer, with impressive survival benefits attained through upregulation of the antitumoral immune response [10]. This checkpoint blockade therapy has mainly focused on treatments targeting surface proteins, programmed cell death 1 (PD-1), its ligand, programmed cell death ligand 1 (PD-L1), and cytotoxic T-lymphocyte-associated protein 4 (CTLA-4), that have resulted in tumor regression in certain cancer patients. Both PD-1 and CTLA-4 are proteins expressed on the surface of cytotoxic T-lymphocytes. At present, three engineered humanized immune-checkpoint inhibitors targeting either PD-1 (Pembrolizumab, Nivolumab) or CTLA-4 (Ipilimumab) have been approved by the Food and Drug Administration (FDA) for human patients with advanced melanoma [10]. 

However, blocking regulatory checkpoint molecules can also result in aberrant immune activation leading to undesirable off-target inflammation and autoimmunity. Despite the impressive benefits of the immune-checkpoint blockade, its use can be hampered by the occurrence of serious adverse events, which can affect multiple organs of the body, including skin, the gastrointestinal tract, the kidneys, both peripheral and central nervous systems, liver, lymph nodes, eyes, pancreas, and the endocrine system. They can range from mild to severe and can even be fatal [10]. 

In dogs, a recent pilot clinical study provided the first evidence of clinical benefit in blocking the PD-1/PD-L1 pathway [8]. PD-1 suppresses T-cell activation upon binding to its ligands, PD-L1 and PD-L2. In humans, the expression of PD-1 is upregulated in tumor antigen-specific T-cells. Aberrant PD-L1 expression in tumor cells, and other cells in the tumor microenvironment, has been demonstrated in various cancer types. Blocking this pathway would restore multiple effector functions of antigen-specific T-cells with subsequent remission of cancer [8]. In the aforementioned clinical trial, a rat–dog chimeric anti-PD-L1 mAb was prepared and the clinical efficacy in seven dogs with oral malignant melanoma (OMM) was evaluated [8]. The chimeric mAb seemed to be safe and well tolerated. The objective response rate (complete and partial response according to the veterinary Response Evaluation Criteria in Solid Tumors (RECIST) V1.0 guidelines) [11] of dogs with OMM was 14.3% (1/7). This dog had stage II disease (primary tumor 2–4 cm in diameter, no involvement of lymph nodes) [12] (Table 1) and an 81% reduction of the tumor burden was observed. In the other four dogs with OMM and confirmed pulmonary metastasis, the estimated MST was superior (93.5 days) compared with a historical group from the same hospital (54 days) [8].

### 2.2. Antiganglioside Monoclonal Antibodies

Human tumors originating from neuroectodermal cells, such as malignant melanoma, express high levels of disialogangliosides GD2 and GD3, making these antigens ideal targets for mAbs. In a study, the expression of disialogangliosides GD2 and GD3 on canine oral malignant melanomas and their in vitro ability to mediate antibody-dependent cellular cytotoxicity (ADCC) using murine anti-GD2 and -GD3 mAbs was assessed. The data from this study indicated that disialogangliosides GD2 and GD3 were expressed in fresh canine melanoma cells. In addition, mAbs reacted with these antigens and could target and trigger tumor killing by multiple canine effector populations. An additional effect when combined with interleukin (IL)-2 was also observed [13]. In another study, the in vitro cytotoxic potential of fresh and IFN-gamma-activated pulmonary alveolar macrophages from normal dogs targeted to canine malignant melanoma cells with antiganglioside mAbs was evaluated. Antiganglioside mAbs significantly enhanced the cytotoxicity of canine melanoma mediated by canine pulmonary alveolar macrophages. Furthermore, melanoma cytotoxicity was enhanced when combined with recombinant canine IFN-gamma-activated canine pulmonary alveolar macrophages. This strategy could provide a potential defense against cancer cells metastatic to the lung [14]. 

## 3. Nonspecific Immunotherapy Activated by Bacteria 

Signature molecules released by bacteria termed pathogen-associated molecular patterns (PAMPs) activate innate immune responses (nonspecific) through the activation of so-called pattern recognition receptors (PRRs) and downstream cellular pro-inflammatory signaling [15]. Important examples of such receptors include Toll-like receptors (TLRs) and a cytoplasmic receptor known as nucleotide-binding oligomerization domain (NOD)-like receptors (NLRs). Both TLRs and NLRs activate downstream pro-inflammatory signaling through nuclear factor-κB (NF-κB) and mitogen-activated protein kinases (MAPK) pathways [16]. This approach has been used in tumor immunotherapy in veterinary oncology. One study evaluated the combination of surgical excision with the intratumoral administration of a heat-killed suspension of *Corynebacterium parvum* to treat dogs with OMM. A longer survival time was observed in dogs with stage II and III diseases, suggesting that *C. parvum*, when combined with surgery, may have antitumor activity in the canine melanoma model [17].

A randomized, double-blinded clinical trial evaluated the efficacy of a liposome-encapsulated lipophilic derivative of muramyl dipeptide (L-MTP-PE) when used in a surgical adjuvant setting and administered alone or in combination with recombinant canine granulocyte macrophage colony-stimulating factor (GM-CSF) to treat 98 dogs with spontaneous OMM. In rodents, dogs, and humans, it was proven that L-MTP-PE, when administered in vivo, could activate monocytes and macrophages, resulting in antitumor activity. In this study, the L-MTP-PE administered alone and in combination had minimal antitumor activity. The study could not demonstrate any therapeutic advantage of GM-CSF over L-MTP-PE alone. However, there was suggestive evidence that L-MTP-PE resulted in the prolongation of survival in dogs with early (stage I) OMM [18]. 

## 4. Oncolytic Virotherapy

Oncolytic virotherapy is an emerging approach to treat cancer. However, oncolytic virotherapy in veterinary medicine is still in its very early stages. Antitumor mechanisms of oncolytic viruses are not fully elucidated, but they can selectively infect, replicate, and lyse tumor cells, and promote immune antitumor responses by different mechanisms [19,20]. Different viruses have been tested for canine cancer therapy in murine models and canine tumor cell cultures, for example, adenovirus, poxvirus, reovirus, vesicular stomatitis virus, and paramyxovirus [19]. Most viruses are genetically modified to selectively lyse tumor cells without affecting normal cells, while some viruses have a natural phenotype which allows them to replicate only in cancer cells [21,22]. Several limitations of this approach include selective targeting of the oncolytic viruses to tumor tissue, relatively poor virus-spreading throughout solid tumor tissue, inefficient viral replication in immune-competent hosts, and disadvantageous ratio between anti-viral and anti-tumoral immunity. Some strategies to improve oncolytic virotherapy efficacy are currently being investigated, such as the improvement of oncolytic vector systems and the combination of oncolytic viruses with conventional cancer therapies [19]. Two studies have analyzed the oncolytic potential of selected viruses against canine cancer cells, including malignant melanoma, in cell cultures [21,23]. One study investigated the oncolytic effect of a strain of naturally oncolytic reovirus (Dearing strain of reovirus serotype 3) on six CMM cell lines. The cells were susceptible to reovirus in a multiplicity of infection (MOI)-dependent manner. At an MOI of 1.000, all (n = 6) CMM cell lines were highly susceptible, whereas at an MOI of 70, 20–50% of cell death was observed in three cell lines, and more than 50% in one cell line [21]. In another study, they analyzed the oncolytic potential of a genetically manipulated Lister strain of vaccinia virus (LIVP6.1.1) against four different canine cancer cells in cell culture, one of them a canine melanoma cell line, and xenografts in nude mice. The LIVP6.1.1 virus was highly cytotoxic to three cell lines, including the melanoma one, resulting in at least 83% cytotoxicity after three days of virus infection at an MOI of one [23]. One dog with melanoma was treated with toceranib and intravenous weekly administration of an upgraded canine version of Celyvir (dCelyvir®), using dog mesenchymal stem cells and ICOCAV17, a new canine oncolytic adenovirus, at an MOI of one [24]. This dog had stable disease [11,24]. Finally, a modified version of the oncolytic vesicular stomatitis virus, VSV-GP, in which the vesicular stomatitis virus glycoprotein G was substituted with the lymphocytic choriomeningitis virus glycoprotein GP, was studied in several models of malignant melanoma [25]. VSV-GP has several strengths, such as a fast replication cycle, no pre-existing immunity in the general population, and the capability to accommodate immunostimulatory cytokines or tumor antigens. One dog, one mouse, and a panel of human melanoma cell lines in vitro were used. In addition, they studied the number of lung micro-metastases in a syngeneic lung metastasis in a mouse model. VSV-GP efficiently lysed more than 50% of cells of all cell lines at an MOI of 0.1 after 48 hours, prolonged the survival of mice and the number of lung metastasis was significantly reduced in treated mice [25]. 

## 5. Vaccines

The discovery of tumor-associated antigens (TAAs) has allowed for the development of techniques to specifically target neoplasms immunologically [26]. Therapeutic vaccination strategies against cancer are based on the concept that cancer cells display a gene (and protein) expression pattern significantly different from their natural counterparts. Vaccination aims to educate the immune system to recognize components of this novel signature. Proteins differentially expressed by cancer cells and capable of inducing effector immune responses are generically defined as TAAs [27]. Multiple vaccination strategies have been devised to induce an antitumor immune response, including dendritic cell (DC) vaccination, and allogeneic whole-cell tumor vaccines [28]. DCs are the most powerful antigen-presenting cells and are considered to link the innate and adaptive immune systems. Thus, DC vaccines are a very powerful antitumor immunotherapeutic strategy. The main goal is the activation of an immune response able to eliminate cancer cells and to produce long-lasting immunity. DC vaccines make use of DC precursors, differentiation into DCs, and loading with relevant TAAs. There is no standardized procedure, which results in a plethora of methods differing in the source of DCs, the nature and procedure for antigen loading, the maturation stimulus, and the route of administration. DCs can be loaded with peptides, proteins, and tumor lysates, viral vectors can be used to insert genes into DCs, encoding TAAs or proteins, and they can even be transfected with mRNA-encoding TAAs [27]. Using this strategy, autologous DCs derived from bone marrow were ex vivo expanded and transfected with an adenovirus expressing the human melanoma antigen gp100. The vaccine was administered subcutaneously in three dogs: One healthy, and the other two with stage I and III oral melanoma after surgical excision and radiotherapy. No adverse effects were observed. Two of the vaccinated dogs did not experience recurrent disease at the site of the initial lesion, nor at distant sites 48 and 22 months after the DC vaccination, respectively [29]. In another study, in vivo evidence of T-cell mediated immunity was demonstrated by delayed-type hypersensitivity skin testing in three healthy dogs, following vaccination with autologous DC cells that were pulsed with lysates from a canine malignant melanoma cell line (CMM-2). Recruitment of CD8 and CD4 T-cells was detected in the positively responding sites, suggesting that the vaccination efficiently elicited a T-cell-mediated immunity against CMM-2 cells in vivo [30]. However, optimization of these vaccination regimes is needed to cement their place in the veterinary cancer clinic [28]. 

Whole-cell tumor vaccines are another approach. Various protocols can be used to prepare whole tumor cells for patient vaccination [28]. Whole-cell tumor vaccines can be administered as irradiated tumor cells or lysed tumor cells, most often with a vaccine adjuvant. The whole tumor cell vaccine approach provides the ability to generate an immune response against a large number of potential tumor antigens, especially in the case of autologous tumor cell vaccines. Drawbacks to whole tumor cell vaccines include the time and expense associated with vaccine preparation and the need for established tumor cell lines. The approach is not readily scalable for mass production and quality control is a major issue [31]. In a phase II clinical trial using allogeneic whole-cell vaccination, a canine melanoma cell line was transfected with xenogeneic human gp100, killed by irradiation and administered intradermally to 34 dogs with spontaneous malignant melanoma (stages II to IV) in an attempt to break tolerance with a combination of self and xenogeneic antigens. The vaccine was well tolerated in dogs [32]. Objective evidence of tumor regression (one complete response and five partial responses) [12] was observed in six of the 34 dogs (17.6% objective response rate). Additionally, dogs experiencing tumor control survived significantly longer (337 days) than dogs having no response (95 days) [32].

## 6. Gene Therapy 

Gene therapy relies on the delivery of foreign DNA into cells also known as transfection. The transfer of DNA can be accomplished by non-viral vectors utilizing liposome delivery or DNA protein complexes, and by viral vectors. Viral vectors are genetically modified viruses, which are still able to transfer their genetic material to a host-cell [33]. There are different gene products that can be delivered, such as cytokines, suicide genes, tumor or bacterial antigens, and proapoptotic genes [34]. 

Cytokines are able to promote adaptive as well as innate immune responses [35]. Intratumoral delivery of cytokines may induce antitumor responses without the adverse effects often associated with systemic delivery [34]. Several cytokines have been evaluated in CMM. IL-2 is a T-cell growth factor that has documented efficacy against human melanoma and melanoma in murine models [35], and has been studied in naturally occurring CMM, combined with tumor surgery and radiotherapy. Recombinant human IL-2 was administered with repeated local injections of xenogeneic Vero cells secreting high levels of this cytokine. The study showed that dogs relapsed less frequently and survived longer (MST: 270 days) than control animals (75 days) treated by surgery and radiotherapy alone [36]. IL-12 has been studied in several papers either as human or feline recombinant [37,38,39], delivered with plasmids or naked plasmids. IL-12 has different immune regulatory roles including the activation of cytotoxic T-lymphocytes and natural killer cells, the production of interferon-gamma, antiangiogenic properties, and inducing apoptosis in tumor cells [40]. In these studies, minimal side effects and clinical partial responses [11] were observed. However, a small number of patients (n = 3) were included, no control group was established, patients had progressive disease [11], and IL-12 was usually combined with chemotherapy [37,38,39].

Superantigens are bacterial proteins that, due to their unique structure, are capable of activating a large number of T-cells [34]. However, systemic exposure to superantigens can be associated with serious toxicity and potential for the induction of T-cell anergy. The local expression of superantigens via gene delivery may be able to effectively activate tumor-infiltrating immune effector cells, while avoiding the adverse effects associated with systemic exposure [41]. This approach has been used in CMM [41,42]. One report evaluated bacterial superantigens and lipid-complexed plasmid DNA encoding staphylococcal enterotoxin B combined with either growth factor GM-CSF or IL-2 [42]. Another study used lipid-complexed plasmid DNA encoding staphylococcal enterotoxin A combined with canine IL-2 [41]. In the former study, the median survivals ranged from 427 days in dogs (n = 3) with stage I disease to 168 days in stage III dogs (n = 12), significantly higher than historical surgery controls (105 days). An overall response rate of 46% was observed [42]. The latter study reported an overall decrease in tumor size in 25% of the dogs (n = 16) with different tumor types, but only two dogs with melanoma were included and the response in these dogs was not specifically reported [41]. In these studies, the toxicity was generally minimal and consisted mainly of transient fever and diarrhea [1]. 

The induction of apoptosis has been shown to enhance immune responses against tumor antigens. There is evidence that various pathways that promote intrinsic apoptosis are commonly inactivated in canine melanoma, and have been evaluated for OMM. Intratumoral administration of the human Fas-ligand gene was evaluated in a phase I clinical trial in four dogs with stage III disease. In three dogs, a 12.5–58% reduction of tumor burden was reported, and no adverse effects were observed in all dogs [43].

Suicide gene therapy, also called gene-directed enzyme prodrug therapy, relies on the capacity of the gene product to convert a non-toxic prodrug into a toxic compound [33]. This approach has been evaluated by Finnochiaro and colleagues for CMM using intratumoral injections of lipid-complexed plasmid DNA encoding the herpes simplex thymidine kinase suicide gene that sensitizes transfected cells to ganciclovir. The suicide gene therapy surgery adjuvant was combined with irradiated xenogeneic cells genetically modified to secrete human IL-2 and GM-CSF [44,45], and with an autologous or allogeneic formolized tumor cell vaccine [45] plus interferon-β [46]. The patients treated with the combination had significantly longer overall survival than the controls (untreated or surgery alone or suicide gene therapy alone) with minimal toxicity. Taking the data from the three studies long-term, the safety and efficacy of this treatment are supported by the high number of treated dogs (n = 629) [44,45,46] with >50% of those dogs dying from non-melanoma-related causes after extensive follow-up (nine years) [47]. However, it is difficult to draw conclusions from these studies [44,45,46,47] as suicide gene therapy was combined with more than one intervention without control groups.

Another immunogene therapy approach is the adjuvant use of a xenogeneic antigen DNA vaccination encoded by a bacterial plasmid. Xenoantigens reported to treat canine OMM are the chondroitin sulfate proteoglycan-4 (CSPG4) and the human tyrosinase [40] genes. CSPG4 is an early cell surface progression marker associated with tumor cell migration, invasion and proliferation. Two trials reported the used of CSPG4 [48,49]. In a first trial [48], the immunogenicity, safety, and therapeutic efficacy were evaluated [40,49]. Dogs with stage II and III surgically resected CSPG4-positive oral malignant melanoma were subjected to monthly intramuscular plasmid administration, which was followed immediately by electroporation (electrovaccination) to increase immunogenicity from six to 20 months. Survival times were significantly longer in 14 vaccinated dogs, compared with 13 non-vaccinated controls [28,48]. In a second trial [49], the same protocol was performed in 23 dogs with surgically excised CSPG4-positive oral canine melanoma. In parallel, 19 control dogs with CSPG4-positive tumors were subjected only to surgery. Vaccinated dogs displayed an MST of 684 days, whereas non-vaccinated dogs showed an MST of 200 days [40,49]. In both trials, no clinically relevant local or systemic side effects were found. These results suggested that xenogeneic electrovaccination against CSPG4 seemed to be effective in treating canine malignant melanoma [28,40,48,49,50].

In February 2010, OnceptTM was the first cancer vaccine to receive full approval from the US Department of Agriculture. On the other hand, the company withdrew the application for the European Medicine Agency in 2014 [50]. However, this vaccine is the only approved therapeutic treatment for OMM. OnceptTM is a bacterial plasmid DNA vaccine encoding the human tyrosinase gene and is licensed for the adjuvant treatment of stage II and III OMM after loco-regional control [28]. The vaccine is administered with a needle-free device, intramuscularly. One paper reported that vaccine delivery via microseeding was safe and feasible [51]. After approval, several studies have reported efficacy with contradictory results [52,53,54,55,56,57]. Some papers reported the vaccine to significantly prolong survival in CMM [52,55], whereas others did not [53,54,56,57]. Inclusion criteria, tumor localization and staging, and treatment protocols were heterogeneous across the studies. MSTs from these studies ranged from 335 [56] to 477 days [53]. The quality of evidence in these studies is low due to either low numbers of dogs included [53,55], the retrospective nature [53,54,55,56,57], and level of censoring (more than 50% of vaccinated dogs were censored from the analysis) [52]. In addition, for these reasons, two papers [53,54] were recently questioned in an editorial [58]. At the present time, the systematic use of Oncept^TM^ to treat CMM cannot be recommended with confidence. The same approach with xenogeneic murine instead of the human tyrosinase gene has been studied in dogs with digit melanoma. In this study, the authors concluded that the vaccine was safe and appeared effective as an adjunctive treatment. An MST of 476 days was reported [59]. 

Cluster of differentiation 40 (CD40) is a co-stimulatory molecule (secondary signal) found on the surface of B-cells and antigen-presenting cells (APCs). When it binds to CD40 ligand (CD40L) expressed on T-cells, it activates the CD40 bearing cells and thus enhances humoral as well as cell-mediated immunity [28]. CD40 also induces apoptosis in cancer cells [1]. Treatment of CMM with a replicate-deficient adenovirus as a vector, expressing the immunostimulatory gene CD40L (AdCD40L), administered intra- and peritumorally after either incisional or excisional surgery was piloted [1] and succeeded by a larger clinical trial of CMM patients [60]. In the pilot study, one case of advanced stage III oral melanoma, treated by intratumoral vaccination and cytoreductive surgery, did not relapse and survived for 401 days. A second case of conjunctival malignant melanoma, treated only with intralesional injections, showed a continuous remission for more than 150 days. Only reversible minor side effects were reported [1]. A larger clinical trial to treat canine melanoma (14 oral, four cutaneous, and one conjunctival) using the same approach was reported. One to six intratumoral injections of AdCD40L were given every seven days, followed by cytoreductive surgery in nine cases, and only immunotherapy in 10 cases. Median survival was 160 days (range: 20–1141 days), with three dogs still alive at submission [60]. These works demonstrated that local adenovector immunogene therapy using CD40L was safe and could have beneficial effects on dogs [1,60].

## 7. Lymphokine-activated Killer (LAK) Cell Therapy

T lymphocytes are important for anti-tumor immune responses. Therefore, enhancement of T lymphocytes may have utility in slowing, or halting, the progression of malignant tumors [61]. Adoptive cell transfer treatment involves the infusion of tumor-specific T lymphocytes into the circulatory system of cancer patients and has been extensively studied in human oncology. Adoptive cell therapy has not been used in veterinary oncology except for ongoing research in canine lymphoma [7,62]. In contrast to adoptive T-cell transfer therapy, which exploits tumor-specific T-cells, passive immunotherapy, referred to as lymphokine-activated killer (LAK) cell therapy, involves the administration of autologous activated lymphocytes without cancer specificity [7]. These lymphocytes have been previously proliferated by stimulation and culture with certain cytokines [63]. This form of immunotherapy is expected to trigger the cytotoxic activity of the administered lymphocytes against target tissues, and indirectly induce cell-mediated immunity by activation of T lymphocytes and natural killer cells [63]. In human medicine, LAK therapy has been applied for patients with tumors in combination with surgery to prevent postoperative tumor recurrence and metastasis by improving the postoperative immune responses suppressed by surgical stress [61]. Therefore, LAK cell transfer is not suitable as a monotherapy, but its application is promising as an adjuvant treatment [7]. Initially, a paper reported the use of LAK cell therapy in healthy beagles and concluded that it was safe and could promote an immune response [63]. LAK therapy has been evaluated in 15 tumor-bearing dogs, seven of them with malignant melanoma, combined with palliative surgery. T lymphocytes were generated from autologous peripheral blood mononuclear cells by culture with recombinant human IL-2 and solid phase anti-canine CD3 antibodies. Therapy was administered intravenously at two to four-week intervals [61] and resulted in an increase of CD8^+^ T-cells and a decreased ratio of CD4+ to CD8+ T-cells [7]. Although promising results, the sample size was small, only three phenotypes were evaluated (CD3+, CD4+, and CD8+), and no clinical follow-up was performed [61].

## 8. Conclusions

Immunotherapy encompasses a wide range of different treatment modalities (Figure 1) and has evolved enormously in the last decade. It has been used for different cancer types in human and veterinary medicine, including malignant melanoma. It seems a promising therapy for CMM, however available evidence does not support the systematic use of this approach as the number of treated dogs is small, treatment protocols usually used more than one approach, and randomized, prospective, and double-blinded clinical trials are lacking.

## Figures and Tables

**Figure 1 vetsci-06-00015-f001:**
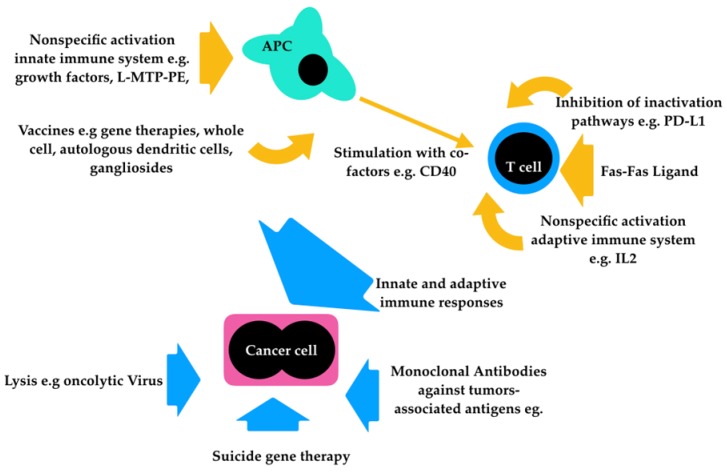
Summary of the different strategies for immunotherapy in CMM.

**Table 1 vetsci-06-00015-t001:** World Health Organization staging system for canine oral malignant melanoma.

Stage I	Stage II	Stage III	Stage IV
≤2 cm diameter	2–4 cm diameter	>4 cm diameter	Any size
No involvement of lymph nodes	No involvement of lymph nodes	+/− metastatic lymph nodes	Distant metastasis

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
