# Peer review of "A Review of Immunotherapeutic Strategies in Canine Malignant Melanoma"

_vetsci, 2019, doi:10.3390/vetsci6010015_

Round 1
Reviewer 1 Report
Authors completely reviewed updated immunotherapeutic strategies for canine malignant melanoma and wrote this easy-reading manuscript. It will greatly benefit veterinarians and researchers who are working on the study of canine melanoma therapy. I recommend to accept it for publication after minor revision listed below.
1. Page 1 Line 7-10 of Introduction: Melanomas involving the haired-skin that are not in proximity to “17” mucosal margins often behave in a benign manner, whereas oral or mucosal melanomas are often malignant and metastatic with reported median survival time (MST) of 14 months for stage I and three months for stage III disease [2]. What is the mean of “in proximity of 17 mucosal margins" ? Wrong typing?
2. Page 6 Paragraph two 5. Vaccines; Please put “(TAAs)” behind “The discovery of tumor-associated antigens” on the first line and delete “(TAAs)” on the 7th line of the same paragraph.
3. Page 8 Paragraph 3; Oncept is a trade name. It should be presented in Oncept™, not OnceptTM.
Author Response
Point 1. Wrong typing. Amended as suggested.
Point 2. Amended as suggested.
Point 3. Amended as suggested.
Reviewer 2 Report
Manuscript ID vetsci-430765
This is a well-written and comprehensive review on immunotherapics that can be used in canine melanoma. I have only some minor comments.
Overall the introduction should be re-organized:
Melanomas involving the haired-skin that are not in proximity to 17 mucosal mar-gins often behave in a benign manner.
What do you mean by this?
Please expand the prognostic significance of each of the mentioned variables (site, size, stage and histologic parameters)
The following sentence As a result newer approaches for systemic management have been developed, for instance, gene therapy [4], or l_y_m_p_h_o_k_i_n_e_‐a_c_t_i_‐vated killer (LAK) cell therapy [5] does not belong here, rather should be moved to the immunotherapy paragraph.
The paragraph:It should be noted that the domestic dog (Canis lupus familiaris) is a useful model for translational medicine in oncology. Canine tumors share with human tumors similar epidemiology, genetic, biology, treatment responses, prognosis factors and clinical outcomes [5]. Therefore, studies may be beneficial not only for developing dog cancer treatment but also to inform human preclinical studies [8] is out of context.
Overall Table 1 is very confusing and should be revised. Also, please add a column specifying whether immunotherapy was used as single treatment modality or in combination with other.
“Conventional treatments”. There are studies with definitely longer MST. Please adjust.
Ref 2 does not describe non-treated dogs
Ref 58 refers to vaccinated dogs.
Ref 29: which one? How can you then describe a median ST if there were only 2 dogs?
Study design: either is listed for all studies or for none.
Pag 6.
This dog had stable disease
For how long?
Author Response
Response to Reviewer 2 Comments
Dear Reviewer,
Thank you for your comments. We have corrected as suggested, herein are the responses:
Point 1: Melanomas involving the haired-skin that are not in proximity to 17 mucosal mar-gins often behave in a benign manner.
What do you mean by this?
Please expand the prognostic significance of each of the mentioned variables (site, size, stage and histologic parameters)
Response 1: Amended as suggested. 17 has been erased (typographical error) and prognostic variables expanded. (See below introduction paragraph)
Point 2: The following sentence As a result newer approaches for systemic management have been developed, for instance, gene therapy [4], or l_y_m_p_h_o_k_i_n_e_‐a_c_t_i_‐vated killer (LAK) cell therapy [5] does not belong here, rather should be moved to the immunotherapy paragraph.
Response 2: Amended as suggested. (See below introduction paragraph)
Point 3: The paragraph:It should be noted that the domestic dog (Canis lupus familiaris) is a useful model for translational medicine in oncology. Canine tumors share with human tumors similar epidemiology, genetic, biology, treatment responses, prognosis factors and clinical outcomes [5]. Therefore, studies may be beneficial not only for developing dog cancer treatment but also to inform human preclinical studies [8] is out of context.
Response 3: Amended as suggested.(See below introduction paragraph)
Point 4: Overall Table 1 is very confusing and should be revised. Also, please add a column specifying whether immunotherapy was used as single treatment modality or in combination with other.
“Conventional treatments”. There are studies with definitely longer MST. Please adjust.
Ref 2 does not describe non-treated dogs
Ref 58 refers to vaccinated dogs.
Ref 29: which one? How can you then describe a median ST if there were only 2 dogs?
Study design: either is listed for all studies or for none.
Response 4: The authors agree, it is very confusing and does not add value, thus the table has been removed.
Point 5: Pag 6.
This dog had stable disease
For how long?
Response 6: It is not specified in the article. In the text the authors refer the reader to the table. See patient UAX-16 in table 1. SD = stable disease.
1. Introduction
The melanoma is a relatively common cancer of dogs arising from melanocytes and accounting for 3% of all neoplasms and up to 7% of all malignant tumor and the most common oral malignancy (56%). Primary melanomas can also occur in the nail bed, footpad, eye, gastrointestinal tract, or mucocutaneous junction [1,2]. The biologic behavior of canine malignant melanoma is extremely variable and best characterized based on anatomic site, size, stage, and histologic parameters. The anatomic site of melanoma is highly, although not completely, predictive of local invasiveness and metastatic propensity. Melanomas involving the haired-skin that are not in proximity to mucosal margins often behave in a benign manner, whereas oral or mucosal melanomas are often malignant and metastatic with reported median survival time (MST) of 14 months for stage I and three months for stage III disease. Size is evaluated in the staging of dogs with melanoma along with lymph node involvement and distant metastasis. For instance, tumors <2 cm in diameter and without evidence of metastasis are classified as stage I with reported MST up to 19 months. On the other hand, tumors between 2-4 cm in diameter without evidence of metastasis are classified as stage II and have reported MST up to 6 months (Table 1). Histologic parameters such as Ki67 expression, a proliferation index histologic marker, can also predict behavior as in canine cutaneous melanoma. An increased Ki67 expression correlates with a moderate-to-high metastatic propensity [2]. Classical clinical management includes radical surgical excision [3], radiation therapy either as primary or adjuvant therapy, and combination of chemotherapy and radiation therapy [2]. These classical therapies remain a challenge to control disease.
It is increasingly recognized that the immune system plays critical roles in the development and progression of cancer [4]. The treatment of cancer has largely relied on killing tumor cells with chemotherapy and radiotherapy. This approach, however, has limitations including severe systemic toxicities, bystander effects on normal cells, recurrence of drug-resistant tumor cells, and the inability to target micrometastases or subclinical disease [5]. In addition, there is no clear evidence that chemotherapy extends MSTs and decreases local recurrence or metastasis risks [2]. Thus, the pursuit of more efficacious and tolerable treatments, in conjunction with an increased understanding of cancer immunology, spurred intense research in the development of immunotherapies directed against cancer [5]. These immunotherapies have emerged as an interesting approach to manage canine malignant melanoma (CMM) and multiple immunotherapeutic strategies have been developed targeting the innate and/or adaptive arms of the immune system. As a result newer approaches for systemic management have been developed, for instance, gene therapy [6], lymphokine‐activated killer (LAK) cell therapy [7], or monoclonal antibodies [8]. These immunotherapies seem in some cases to improve survival times.
The aim of this review is to summarized current most relevant immunotherapeutic strategies for the treatment of CMM divided among five different categories: monoclonal antibodies, nonspecific immunotherapy activated by bacteria, vaccines, gene therapy and lymphokine-activated killer cell therapy. Studies performing at least two approaches fell in one category choosed at the discretion of the author. This review focuses on most relevant English written, peer-reviewed papers of preclinical experiments and clinical data collected from client-owned dogs with spontaneously arising tumors. The search was based on PubMed, Web of Science and Scopus using the Boolean chains “dog” OR “dogs” OR “canine” AND “melanoma” AND “immunotherapy” OR “vaccine”. Search was performed and revised within a time frame from October the first until December the 15th of 2018, and retrieved a total of 241 papers. Irrelevant, duplicated or out of beyond the scope papers were excluded. Sixty-four papers were included for the review.
